# Genetic Risk Factors for Neurological Disorders in Children with Adverse Events following Immunization: A Descriptive Study of a Polish Case Series

**DOI:** 10.3390/ijms24021117

**Published:** 2023-01-06

**Authors:** Agnieszka Charzewska, Iwona Terczyńska, Agata Lipiec, Tomasz Mazurczak, Paulina Górka-Skoczylas, Róża Szlendak, Karolina Kanabus, Renata Tataj, Mateusz Dawidziuk, Bartosz Wojtaś, Bartłomiej Gielniewski, Jerzy Bal, Elżbieta Stawicka, Dorota Hoffman-Zacharska

**Affiliations:** 1Department of Medical Genetics, Institute of Mother and Child, Kasprzaka 17A, 01-211 Warsaw, Poland; 2Institute of Mother and Child, Clinic of Paediatric Neurology, Kasprzaka 17A, 01-211 Warsaw, Poland; 3Laboratory of Molecular Neurobiology, Nencki Institute of Experimental Biology PAS, Pasteura 3, 02-093 Warsaw, Poland; 4Institute of Genetics and Biotechnology, University of Warsaw, Pawińskiego 5a, 02-106 Warsaw, Poland

**Keywords:** epilepsy, vaccination, drug-related side effects and adverse reactions, exome sequencing

## Abstract

Studies conducted on large populations show a lack of connection between vaccination and serious neurological symptoms. However, there are isolated cases that indicate such a relationship. These reports on adverse effects following immunization (AEFI) reduce social confidence in vaccination; however, their background may be rare genetic defects. The aim of the presented study was to examine if neurological AEFI in children may be associated with variants in genes related to neurodevelopment. To identify such possible associations, a descriptive study of the Polish case series was conducted. We performed next-generation sequencing in patients who, up to 4 weeks of injection of any vaccine, manifested neurological AEFI. We included 23 previously normally developing children with first seizures that occurred after vaccination. We identified pathogenic/likely pathogenic variants in genes engaged in neurodevelopment in nine patients and variants of uncertain significance in another nine patients. The mutated genes belonged to the group of genes related to epilepsy syndromes/epileptic encephalopathy. We showed that AEFI might have a genetic background. We hypothesized that in some AEFI patients, the vaccine might only trigger neurological symptoms that would have been manifested anyway as a result of a pathogenic variant in a gene engaged in neurodevelopment.

## 1. Introduction

Vaccination is one of the most significant advances in medicine in modern times. Many catastrophic diseases, such as black pox and polio, have been successfully eradicated with the introduction of mass vaccination. Black pox was declared eradicated in 1980, and in 2015 and 2019, it was announced that the polio virus wild type II and III, respectively, were eliminated worldwide [1]. As these diseases are completely unknown to modern parents, they are increasingly questioning the need to immunize their children for them, especially since no vaccine is completely free from side effects or the risk of complications [2]. To maintain herd immunity and prevent disease outbreaks, an immunization rate of 95% is required [3]. However, the spread of concern regarding adverse events following immunization (AEFI) is causing the mass refusal of parents to vaccinate their children and a decrease in herd immunity. The consequence is the return of serious diseases, such as measles [4], the extent of which, thanks to vaccination, has already been significantly limited. A systematic review study [5] performed in Sweden showed that the benefits of immunization far outweigh the risk of adverse events because immunization has virtually eliminated morbidity and mortality due to many common diseases among children and adults. Maglione et al. [6], in another systematic review of the safety of vaccines used for routine immunization of U.S. children, showed that some vaccines are associated with serious AEFI; however, these events are extremely rare and must be weighed against the protective benefits that vaccines provide. Recent meta-analyses of large case–control and cohort studies have shown that there is no causal link between vaccination and diseases such as autism spectrum disorders and other developmental disabilities [7,8], multiple sclerosis [9], or deaths [10], although case reports indicating such a relationship are being published. These reports cause a massive decline in confidence in vaccination [11]. However, the immediate cause of such events may be genetic background, the presence of rare genetic defects, not the vaccination itself, which was applied by Lin and He [12], who created an ontological framework (Ontology of Genetic Susceptibility Factors, OGSF) to represent various genetic susceptibility factors annotated from vaccine adverse events genetic association studies. Progress in molecular research has also established that many children previously diagnosed with vaccine-related encephalopathy have had Dravet syndrome, congenital developmental and epileptic encephalopathy caused by variants in the *SCN1A* gene [13]. As shown by Paolo Bellavite in his interesting review of AEFI [14], the nature of vaccine-induced pathology is certainly multifactorial, but the knowledge of genetic susceptibility to adverse effects still emerges, and many genetic disorders have already been associated with AEFI. Recent case reports suggest that neurological AEFI can be indeed associated with genetic factors such as pathogenic variants in genes related to epilepsy [15] or ataxia [16]. In the retrospective study of Dravet patients, it was suggested that the vaccine might trigger the earlier onset of the disease but does not have long-term consequences [17]. It is worth emphasizing that contraindications to the administration of vaccines, even in the group of children with neurological diseases, are very limited, apply only to selected preparations, and are very often temporary. Infectious diseases can worsen a child’s neurological condition and disrupt the rehabilitation process, so abandoning vaccinations is not justified, even though AEFI can occur after vaccination [13].

Neurological symptoms after vaccination are mostly temporary. Persistent neurological disorders following vaccination are extremely rare (with a frequency characteristic for rare diseases). In Poland, AEFI are reported on average with a frequency of 0.05% for vaccinations administered under the Protective Vaccination Program and 0.04% for COVID-19 vaccinations [18]. The most frequently observed neurological reaction is febrile seizures, although it should be noted that they occur very rarely (e.g., after DTP vaccination, 6 seizure events per 100,000 vaccine doses administered, and in the case of MMR, 25-34 seizure episodes per 100,000 doses) [13]. As genetic susceptibility factors may stay behind the etiology of neurological AEFI, there is a need for research to establish the role of genetic variants in these adverse events. Thus, the aim of the presented study was to examine if neurological symptoms in children qualified as AEFI may be related to the occurrence of pathogenic variants in genes related to the development of the nervous system.

## 2. Results

We report on 23 children with AEFI for whom next generation sequencing (NGS) was performed. To identify possible genetic risk factors of AEFI, we analyzed variants in genes engaged in neurodevelopment and associated with different forms of epilepsy/epileptic encephalopathy. We identified pathogenic/likely pathogenic variants in 9/23 (39%) patients, 7 of which were acknowledged as definitely pathogenic due to parent examination. Variants of uncertain significance (VUS) were identified in nine patients (39%). In five patients (22%), no variants that could be responsible for AEFI were found; however, this does not exclude the existence of variants in regulatory and non-coding regions or the presence of copy number changes in these patients, as aCGH studies (array comparative genomic hybridization) were not performed. The results of the NGS study are listed in Table 1, and the clinical characteristics of AEFI patients from the study cohort who were diagnosed with pathogenic/likely pathogenic variants (9 patients) are summed up in Table 2 and described in detail in Appendix A.

The study included children who had seizures from the first day to 4 weeks after vaccination, while epilepsy was diagnosed from 2 weeks to 2.5 years after vaccination, after which an AEFI appeared. None of the children presented intrauterine hypotrophy; three children displayed complications in the fetal and perinatal period. One child was born prematurely, the other at term. Neonatal seizures and febrile seizures were not observed. Two children showed MRI abnormalities. None of the children were diagnosed with a neurological syndrome at the time of study inclusion. Polymorphic seizures were found in the analyzed group; 3/9 experienced status epilepticus or cluster seizures.

Table 1 consists of variants that were acknowledged as pathogenic or potentially pathogenic according to the guidelines of the American College of Medical Genetics and Genomics (ACMG) and are located in genes engaged in neurodevelopment, absent, or with extremely low frequency in gnomAD, with in silico predictions as potentially pathogenic, including a CADD score above 20 (cut-off level). The mutated genes belonged to the group of genes associated with epilepsy syndromes/epileptic encephalopathies. We found two pathogenic de novo missense variants in *PCDH19*, the gene related to developmental and epileptic encephalopathy (DEE), type 9 (DEE-9), c.688G>A, and c.1813T>C in two female patients (9 and 12, respectively) who developed epilepsy, developmental regression, and moderate intellectual disability (ID). In *TSC2,* the gene related to tuberous sclerosis, we identified a de novo deletion resulting in the splice site loss c.5252_5259+19del in Patient 14. The patient was initially diagnosed with West syndrome and interestingly, bilateral, subependymal nodules appeared in the brain MRI performed recently, which confirmed molecular diagnosis. In *SCN8A*, the gene related to DEE-13, and cognitive impairment with cerebellar ataxia, we found a missense variant c.5630A>G in Patient 23, who was diagnosed with epileptic encephalopathy, ID, and cerebellar atrophy. Inheritance of the variant is unknown as parents refused to be tested, but the variant is known and described in the literature as pathogenic. We also found two inherited frameshift variants in *PRRT2*, the gene associated with familial, benign infantile seizures with incomplete penetrance c.119_122dup and c.433del in Patients 7 and 13, respectively. The patients were diagnosed with epilepsy. Patient 7 had transient inhibition of development, and Patient 13 also had behavioral problems.

In subsequent genes, *SLC2A1* and *PPP2CA*, related to ID and epilepsy, we found another de novo, pathogenic variants. In *SLC2A1*, the gene related to GLUT1 deficiency syndrome, we identified a novel missense variant c.74A>G in Patient 19, who was diagnosed with epilepsy, paresis, and dystonia. In *PPP2CA*, the gene related to neurodevelopmental disorder, we identified known missense variant c.794A>G in Patient 21, who was diagnosed with epilepsy, developmental regression, and speech disorder. Finally, in *CLTC*, the gene also related to ID and epilepsy, we identified a missense variant, c.3128G>A, in Patient 16, which is described in the HGMD database as presumably pathogenic, but its inherited character suggests to treat it as a variant of uncertain significance.

## 3. Discussion

We performed an NGS study on 23 previously normally developing children with seizures that first occurred after vaccination. To our knowledge, such high-throughput genetic analyses performed on AEFI patients have never been published. Our results suggest a relationship between AEFI and the occurrence of variants in neurodevelopmental genes, such as *PCDH19*, *TSC2*, *SLC2A1*, *SCN8A*, *PPP2CA*, and *PRRT2*. Most of the genes have never been correlated with AEFI in scientific literature. In our case series, neurological symptoms of the AEFI patients corresponded to the clinical signs described for the genes in which we have identified pathogenic variants. Moreover, our findings allowed us to adjust the therapeutic management of AEFI patients according to the molecular diagnosis we made. For example, variants in the gene encoding protocadherin-19 (*PCDH19*) are the cause of developmental and epileptic encephalopathy, type 9, also known as epilepsy, and mental retardation restricted to females (EFMR). It is an X-linked disorder characterized by seizure onset in infancy and mild to severe intellectual impairment affecting heterozygous females only while transmitting males are unaffected. Refractory epilepsy, caused by *PCDH19* pathogenic variants, is a highly drug-resistant condition, but molecular diagnosis has allowed us to apply Ganaxolone in our patients, a drug whose effectiveness in females with *PCDH19* variants has been recently confirmed in clinical trials [19]. Another example of treatment adjustment as a result of molecular diagnosis is the patient with the *SLC2A1* variant. Pathogenic variants in the *SLC2A1* gene cause Glucose Transporter 1 deficiency and decreased glucose concentration in the central nervous system leading to a variety of severe neurological symptoms, including ID, epilepsy, and dystonia. The patients, due to molecular diagnosis, were subjected to MAD treatment (modified Atkins diet), a high fat, low carbohydrate, moderate protein diet that imitates the metabolic state of fasting to induce ketosis and is recommended for patients with glucose transporter deficiency [20].

Another example is the patient with the *TSC2* variant. The patient was initially diagnosed with West syndrome, but after a brain MRI, it appeared that bilateral, subependymal nodules were present, which confirmed the molecular diagnosis of the *TSC2* variant. Tuberous sclerosis is a disease affecting multiple organ systems, so a multidisciplinary team of medical professionals is required. Reverse phenotyping (from genotype to phenotype) and early diagnosis certainly change the therapeutic approaches and can improve the functioning of the patient. 

We showed that neurological symptoms following immunization might possess genetic background information. Our research is consistent with the previous results described by Damiano et al. [21], who identified *SCN1A* variants in post-vaccination encephalopathies. However, we went a step further in our study; we analyzed not only *SCN1A* but all known genes related to epilepsy, showing that neurological symptoms in children qualified as AEFI may be related to the occurrence of pathogenic variants in genes engaged in the development of the nervous system. In total, 39% of pathogenic/likely pathogenic variants that we obtained in our case series were consistent in the diagnostic yield of whole exome sequencing in genetic disorders, which in postnatal cohorts ranged from 25 to 50% [22]. One has to remember that in such a study, we do not analyze non-coding and regulatory regions as well as copy number changes. Thus, the outcome of 39% is relatively high and suggests the relationship between AEFI and genetic factors. Furthermore, genetic factors are not the only factors responsible for AEFI. The pathogenesis of AEFI is multifactorial, but genetic susceptibility plays an important role.

The descriptive case series study which we present here lets us identify probable genetic risk factors that may be associated with AEFI. Based on our preliminary results, we can generate a hypothesis that in some AEFI patients, the vaccine is only a trigger factor for neurological symptoms that would have been manifested anyway as a result of a pathogenic variant in a gene engaged in neurodevelopment. However, this hypothesis should be verified by a study with a greater strength of evidence, i.e., an analytical cohort study. The study design applied in this work has its limitations, such as a small sample size and bias related to patient inclusion. Patients were recruited from the hospital, while the targeted selection of patients based on AEFI central registries will be more appropriate. Moreover, there may be risk factors other than genetic ones in the etiology of AEFI because it is a multifactorial disorder. Another limitation of a case series study is the absence of a comparative group [23]. Therefore, it cannot be stated whether AEFI is really associated with the gene variant rate unless it can be shown that the group with no AEFI has a different gene variant rate from the cases being studied. Thus, the study cannot be used to prove a causal relationship between AEFI and genetic variants. However, its aim was not to test the hypothesis on the relationship between AEFI and genetic risk factors but to generate such a hypothesis. The hypothesis should be verified in further research. 

In addition to these limitations, the promising studies shown here should definitely be extended and performed on large patient cohorts, which should translate into increased social confidence in the immunization programs. The results of such extensive research could serve to create a database with gene variants related to neurological AEFI that could be developed over the years. The database may contribute to the development of screening programs for neurological AEFI risk factors in the future and personalization of the vaccination program, i.e., exclusion/postponement of vaccination for people predisposed to severe neurological AEFI. 

Our results show that more attention should be paid to the role of genetic factors in the pathogenesis of neurological AEFI and that all patients with such reactions should be referred for genetic examination because the outcome may be of importance in therapeutic management.

## 4. Materials and Methods

We conducted a descriptive case series study of Polish patients in the years 2019–2022 at the Institute of Mother and Child (IMC) in Warsaw, Poland, a children’s hospital which is a research and development unit of the Polish Ministry of Health. The design of the study was ambidirectional, with both retrospective (the analysis of existing medical documentation) and prospective (obtaining material for research, performing genetic tests and their analysis) phases of the study. The patients were collected and investigated during the course of the IMC statutory projects (IMC 510-18-49 and IMC 510-18-25) or as a part of routine genetic diagnostics carried out at the IMC. The research was approved by the Bioethical Committee of the IMC, resolution number 5/2018. Understanding and written informed consent from the children’s parents/legal guardians were obtained for all individuals participating in this study. The study was inspired by patients referred to our clinic with post-vaccination encephalopathy, in whom we identified pathogenic gene variants. We were also inspired by the work of Samuel Berkovic [24], who performed a retrospective study on patients with alleged post-vaccination encephalopathy and identified de novo *SCN1A* variants.

In the study, we included 23 patients who met the following inclusion criteria: had normal psychomotor development prior to vaccination and up to 4 weeks of injection of any vaccine, according to the childhood immunization schedule, manifested neurological AEFI that are administratively defined as encephalopathy, febrile seizures, nonfebrile seizures, paralysis, encephalitis, meningitis, and Guillain–Barré syndrome [25]. Patients’ data were analyzed, and selected patients were subjected to pediatric and neurological verification by several independent doctors. Psychomotor development was assessed on the basis of the revised Denver test (Denver II) [26]. Source data from medical documentation were used to determine AEFI onset relative to vaccination. The time interval of 4 weeks was selected on the basis of the Regulation of the Polish Minister of Health [25]. Only patients with true AEFI were selected for the study. Patients who had neurological symptoms with different backgrounds other than vaccination were excluded from the study. Exclusion criteria were past infections and injuries of the central nervous system. 

The adopted criteria for drug-resistant epilepsy were in line with the current recommendations of the International League Against Epilepsy (ILAE)–revision 2020 [27]. EEG tests were performed on all children in certified electrophysiology laboratories. MRI neuroimaging was performed using the epilepsy protocol in the laboratories of reference centers for epilepsy treatment in Poland, using devices with a power of 1.5 Tesla. Among the analyzed group, antiepileptic treatment included most drugs in accordance with the ILAE guidelines and the summary of product characteristics adjusted to the child’s age, types of seizures, or epilepsy syndrome. In Patient 9, Ganaxolone was used at a dose of 50 mg/kg/day for 12 weeks, improving the reduction of epileptic seizures. Patient 19 was placed on the modified Atkins diet. Until the paper was submitted for publication, none of the children in the study group underwent neurosurgical treatment for epilepsy or implantation of a vagus nerve stimulator (VNS). All patients included in the study remain under the care of multi-specialty reference centers. In the study group, four patients, due to significant disorders associated with developmental epileptic encephalopathy, required pediatric, neurological, rehabilitation, neuropsychological, and speech therapy care. The families of the patients with molecular diagnoses were taken care of by the genetic clinic.

Patients with confirmed neurological AEFI were qualified for genetic testing. DNA of the patients were isolated from peripheral blood according to standard procedures. In order to identify possible associations between AEFI and genetic variants, we performed next-generation sequencing. Exome sequencing was performed in 21 patients, and we included an additional 2 patients in the study who came to our clinic for routine diagnostics of epilepsy/epileptic encephalopathy with the use of targeted panel sequencing. Because the patients had pathogenic variants diagnosed with this panel, they were not referred for WES examination.

Exome sequencing was performed based on the Twist Human Core Exome Plus Kit system (Twist Bioscience) on the NovaSeq6000 platform (2 × 100 bp, Illumina) or with SureSelect Human All Exon v6 Kit (Agilent Technologies) on the HiSeq1500 platform (2 × 150 bp, Illumina). Panel sequencing was performed based on the NimbleGen SeqCap Target Enrichment Kit (Roche) on the MiSeq platform (2 × 150 bp, Illumina), using the custom panel for early infantile epileptic encephalopathy (EIEE) genes (EIEE panel, v.1/2016) designed at the IMC based on OMIM [28], HGMD Professional [29], and EpilepsyGene [30] databases, but also based on previous studies [31,32,33] as well as the results of our own research and data from the internal ICM database. The panel included 49 genes, known in 2016 as the molecular background of EIEE syndromes. Not all genes related to EIEEs listed in OMIM at that time were considered; the genes of recessive inheritance identified only in populations with a high percentage of consanguine marriages were omitted.

The analysis of variants in clinically significant genes was performed using a data processing pipeline based on VEP algorithms developed at the Department of Medical Genetics IMC, human genome version GRCh38/hg38. The quality analysis of the obtained sequences was performed using the IGV 2.7 program (Broad Institute). We analyzed variants in genes engaged in neurodevelopment and associated with different forms of epilepsy/epileptic encephalopathy. The gene lists used for WES analysis and panel analysis are described in Appendix A. The obtained variants were filtered by frequency, segregation, pathogenicity, and functionality. Common variants were filtered out (minor allele frequency MAF < 0.02) based on a database search (gnomAD, dbSNP, internal IMC database). The presence of the variants in the HGMD mutation database (HGMD Professional 2021.4) and ClinVar database was checked. Pathogenicity analysis was performed with the use of bioinformatics tools (CADD, Polyphen, Mutation Taster, Sift, SpliceAI). The ACMG classification of variants [34] was conducted using the Varsome tool [35]. The control group consisted of 125,000 people with excluded childhood neurological diseases, whose exomes were sequenced under the gnomAD Exome Consortium [36], and 100 people with excluded childhood neurological diseases from our internal IMC database. Variants identified in patients in the study group were considered potentially pathogenic if present in the control groups with minor allele frequency MAF < 0.01. Variant verification and segregation analysis were performed by Sanger sequencing. 

## Figures and Tables

**Table 1 ijms-24-01117-t001:** Results of NGS study in a cohort of AEFI patients. Pathogenic/likely pathogenic variants in ACMG classification are bold. ACMG classification was conducted using Varsome (https://varsome.com (accessed on 1 August 2022)). HGMD status refers to HGMD Professional 2021.4 (https://my.qiagendigitalinsights.com (accessed on 25 March 2022)). Indicated variants are those in genes engaged in neurodevelopment, absent, or with extremely low frequency in gnomAD, with in silico predictions as likely pathogenic, including CADD score above 20 (cut-off level). ACMG—American College of Medical Genetics and Genomics; VUS—variant of uncertain significance; EIEE—early infantile epileptic encephalopathy; DEE—developmental and epileptic encephalopathy; NDD—neurodevelopmental disorder.

ID	NGS Study	Gene	Reference Sequence	Variant	Zygosity	Inheritance	ACMG Classification	CADD Score	HGMD Status
1	WES	*FAM111A*	NM_001374864.1	c.1578delC, p.(Pro527Leufs*3)	het	maternal	VUS	-	novel
2	WES	*-*							
3	WES	*CREBBP*	NM_004380.3	c.6956A>T, p.(His2319Leu)	het	paternal	VUS	27.7	novel
4	WES	*GRIN1 CACNA1A*	NM_001185090.2 NM_001127221.1	c.467G>A, p.(Arg156His) c.5784G>A, p.(Met1928Ile)	het het	maternal maternal	VUS VUS	25.7 20.9	novel novel
5	WES	*SCN3A*	NM_006922.4	c.4487T>C, p.(Met1496Thr)	het	paternal	VUS	23.5	novel
6	WES	*-*							
**7**	**WES**	** *PRRT2* **	**NM_145239.2**	**c.119_122dup, p.(Pro42Glyfs*93)**	**het**	**paternal**	**Likely Pathogenic**	**-**	**novel**
8	WES	*CACNA1H*	NM_001005407.1	c.2722C>A, p.(Leu908Met)	het	maternal	VUS	25.2	novel
**9**	**WES**	** *PCDH19* **	**NM_001105243.2**	**c.688G>A, p.(Asp230Asn)**	**het**	**de novo**	**Pathogenic**	**26.6**	**known, DEE**
10	WES	*-*							
11	WES	*-*							
**12**	**EIEE panel**	** *PCDH19* **	**NM_001184880.1**	**c.1813T>C, p.(Tyr605His)**	**het**	**de novo**	**Pathogenic**	**27.8**	**novel**
**13**	**WES**	** *PRRT2* **	**NM_145239.2**	**c.433del, p.(Arg145Glyfs*31)**	**het**	**maternal**	**Likely pathogenic**	**-**	**known, Infantile convulsions**
**14**	**WES**	** *TSC2* **	**NM_000548.5**	**c.5252_5259+19del, p.(Arg1751Hisfs*21)**	**het**	**de novo**	**Pathogenic**	**-**	**known, Tuberous sclerosis**
15	WES	*GABRA5*	NM_000810.3	c.896C>T, p.(Thr299Ile)	het	maternal	VUS	26.2	novel
**16**	**WES**	** *CLTC* ** *TRRAP* *CHRNA2*	**NM_001288653.1** NM_001244580.1 NM_000742.3	**c.3128G>A, p.(Arg1043His)** c.818A>G, p.(Asn273Ser) c.325G>A, p.(Val109Ile)	**het** het het	**paternal** paternal paternal	**Likely pathogenic** VUS VUS	**32** 24.1 34	**known, NDD?** novel novel
17	WES	*RELN* *PACS2*	NM_005045.3 NM_001100913.2	c.2677T>C, p.(Tyr893His) c.622T>G, p.(Ser208Ala)	het het	maternal (mother affected)	VUS VUS	24.4 24.0	novel novel
18	WES	*-*							
**19**	**EIEE panel**	** *SLC2A1* **	**NM_006516.2**	**c.74A>G, p.(Gln25Arg)**	**het**	**de novo**	**Likely pathogenic**	**27.0**	**novel**
20	WES	*IFIH1*	NM_022168.4	c.511A>C, p.(Ile171Leu)	het	paternal	VUS	23.4	novel
**21**	**WES**	** *PPP2CA* **	**NM_002715.4**	**c.794A>G, p.(Tyr265Cys)**	**het**	**de novo**	**Pathogenic**	**31**	**known, NDD**
22	WES	*TSC1*	NM_000368.5	c.1364C>T, p.(Thr455Ile)	het	maternal	VUS	24.9	novel
**23**	**WES**	** *SCN8A* **	**NM_014191.4**	**c.5630A>G, p.(Asn1877Ser)**	**het**	**unknown**	**Pathogenic**	**25.7**	**known, Epilepsy and DEE**

**Table 2 ijms-24-01117-t002:** Adverse events following immunization (AEFI) in patients from the study cohort who were diagnosed with pathogenic/likely pathogenic variants in genes engaged in neurodevelopment (Table 1). Hexa-hexavalent vaccine (DTaP-IPV-Hib-HepB); Penta-pentavalent vaccine (DTaP-IPV-Hib); PCV13—Pneumococcal conjugate vaccine, 13-valent; PCV10—Pneumococcal conjugate vaccine, 10-valent; RV—Rotavirus vaccine; MMRV—measles-mumps-rubella vaccine. AED—antiepileptic drug; ID—intellectual disability; ASD—autism spectrum disorder.

ID	Sex	Age (Years)	AEFI	Time after Vaccination	Seizures/Epilepsy	Other Clinical Data
**7**	M	3	(1) 6th week (Penta, RV, PCV10) (2) 5th month of life (Penta)	(1) 2nd day after vaccination (2) 2nd day after vaccination	(2) 10 days after vaccine, tonic seizures, convulsive status epilepticus (tonic-clonic seizures), epilepsy treated with AEDs	transient inhibition of development, hypotonia (1) fever for 2 days, increased muscle tension, bending position, (2) local reaction after vaccination (redness), vaccine was discontinued
**9**	F	3	2nd month of life (PCV13)	4 weeks after vaccination	polymorphic seizures occurring in clusters, refractory epilepsy treated with AEDs	local reaction after vaccination—erythema, parasomnias a month after vaccination, developmental regress, moderate ID, developmental encephalopathy with drug-resistant epilepsy, infection-related clusters seizures, treated with AEDs and Ganoxolone, vaccine was discontinued
**12**	F	4	13th month of life (MMRV)	from 7th day after vaccination	polymorphic seizures: bilateral tonic–clonic seizures, focal motor seizures with impaired awareness occurring in clusters, refractory epilepsy treated with AEDs from the age of 14 months	fever for 24 h, post-vaccination measles rash followed for 7 days, developmental regress, moderate ID, ASD, developmental encephalopathy with drug-resistant epilepsy, infection-related clusters seizures, treated with AEDs and steroidotherapy (Metylprednisolon), vaccine was discontinued
**13**	F	4	(1) 2.5th month of life (Hexa, RV, PCV10) (2) 4.5th month of life (Hexa, RV, PCV10)	(1) 1st day after vaccination (2) 1st day after vaccination	(2) 3 weeks after vaccine (5 months of life), bilateral tonic–clonic seizures of unknown onset and focal seizures with impaired awareness occurring in clusters, diagnosed with epilepsy, treated with AED	suspicion of ASD (1) fever for 24 h, (2) fever for 48 h, anxiety disorders, vaccine was discontinued
**14**	F	4	6th month of life (PCV10)	one week after vaccination	onset of infantile spasms: in the beginning, only single events during wake-up stage, then in the 1.5th month, in all behavioral stages, in series up to 100. Diagnosed with West Syndrome, treated with AED and ACTH	perinatal complication: first, dizygotic dichorionic twin pregnancy (upper respiratory tract infection in the 1st trimester, arterial hypertension), delivery by caesarean section at 36 weeks, birth weight 2350 g (10–50c), 9 AS, developmental regress, moderate ID, speech disorder, vaccine was discontinued. MRI findings: bilateral, subependymal nodules (diameter approximately 4mm), thin corpus callosum, delayed myelination, enlargement of the subarachnoid space, vaccine was discontinued
**16**	F	4	2.5th month of life (Hexa, RV)	3 weeks after vaccination	focal clonic seizures with impaired awareness and bilateral tonic–clonic seizures of unknown onset, infantile spasms during wake up stage, in series up to 60, diagnosed with West Syndrome, treated with AEDs and ACTH	perinatal complication: fetal arrhythmia between 31/32 weeks of gestational age, delivery by caesarean section at 39 weeks, transient tachypnea after delivery, apnea, transient inhibition of development, developmental aphasia, vaccine was discontinued
**19**	F	10	17th month of life (Hexa)	72 h after vaccination	myoclonic seizures, absence seizures, diagnosed with epilepsy from 20th month of life, treated with AED	family history: febrile seizures in mother, father, and mother’s cousins. Apnea, lower limbs paresis, paroxysmal dystonia, consciousness disturbances, early onset absence epilepsy, myoclonic jerks that worsen during an infection with or without fever, treated with AED and MAD (modified Atkins diet), vaccine was discontinued
**21**	M	4	6th month of life (Hexa, PCV13)	1st day after vaccination	focal seizures with impaired awareness, bilateral tonic–clonic seizures after two months	transient developmental regress, speech disorder, fever for 72 h, epilepsy treated with AED from 3 years of age, vaccine was discontinued
**23**	F	12	5th month of life (DTaP, Hib, OPV)	2nd day after vaccination	tonic seizures, focal seizures with impaired awareness, epilepsy treated with AEDs	perinatal complication: moderate asphyxia (5-7-8 AS), apnea, development regress, moderate/severe ID, developmental and epileptic encephalopathy, MRI findings: cerebellar atrophy, vaccine was discontinued except Hepatitis B vaccine

## Data Availability

The data presented in this study are available on request from the corresponding author.

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
