# Peer review of "Genetic Risk Factors for Neurological Disorders in Children with Adverse Events Following Immunization: A Descriptive Study of a Polish Case Series"

_ijms, 2023, doi:10.3390/ijms24021117_

Round 1

Reviewer 1 Report

The paper focused on discussing the adverse effects in children after immunization which explored the role of genetic factors in neurological disorders of these children. This paper elucidates another possibility behind the adverse effects after immunization which is related to the rare genetic defects.  

In this study, Dr. Charzewska and colleagues provided a detailed analysis about the AEFI and the occurrence of variant in neurodevelopmental genes. Identifying those variants would help adjust therapeutic management of AEFI patients according to their genetic profile.

However, there were only 9 out of 23 patients carried pathogenic/likely pathogenic variants which only counts for 39% of total samples. However, for 5 out of 23 patient (22%), there were no identified variants that could be responsible for the AEFI. These results indicate that genetic variants are not the only reason for the neurological symptoms in children with AEFI.  Also, in the table 1, they also showed some of those variants were variants of uncertain significance. In order to support their hypothesis, better explanation needs to be provided.  

Besides, I have some minor suggestions. If the author could spell out all words when they first used abbreviations, such as aCGH, that would be helpful.

Author Response

Reviewer 1_Review Report (Round 1)

The paper focused on discussing the adverse effects in children after immunization which explored the role of genetic factors in neurological disorders of these children. This paper elucidates another possibility behind the adverse effects after immunization which is related to the rare genetic defects.  

In this study, Dr. Charzewska and colleagues provided a detailed analysis about the AEFI and the occurrence of variant in neurodevelopmental genes. Identifying those variants would help adjust therapeutic management of AEFI patients according to their genetic profile.

However, there were only 9 out of 23 patients carried pathogenic/likely pathogenic variants which only counts for 39% of total samples. However, for 5 out of 23 patient (22%), there were no identified variants that could be responsible for the AEFI. These results indicate that genetic variants are not the only reason for the neurological symptoms in children with AEFI.  Also, in the table 1, they also showed some of those variants were variants of uncertain significance. In order to support their hypothesis, better explanation needs to be provided.  

AR: Author’s response: A result of 39% is consistent in the diagnostic yield (25-50 %) of whole exome sequencing in genetic disorders. Please remember that in such study we do not analyze non-coding and regulatory regions as well as copy number changes. Thus, the outcome of 39% is relatively high and suggests the relationship between AEFI and genetic factors. Besides, obviously genetic factors are not the only one, which are responsible for AEFI. The pathogenesis of AEFI is multifactorial, but genetic susceptibility plays an important role. Variant of uncertain significance are those, which have good bioinformatic predictions, but cannot be confirmed as the cause of the disease or refuted based on current data.

Besides, I have some minor suggestions. If the author could spell out all words when they first used abbreviations, such as aCGH, that would be helpful.

AR: Done

Author Response

Reviewer 2_Review Report (Round 1)

Firstly, I would like to congratulate the authors on their interesting and relevant work.
However, there are some points that need to be addressed:

1) The title should contain the study design type.

Author’s response (AR): The title has been changed to:

Genetic risk factors for neurological disorders in children with adverse events following immunization. A descriptive study of Polish case series.

2) The specific study design type should also be mentioned in the abstract.

AR: Done

3) I advise the authors to include standardized key-words according to MeSH terms
as this facilitates the search of the article in the future. Please see this website for
more information: https://www.ncbi.nlm.nih.gov/mesh/

AR: Done

4) L39: The authors should adequate the references according to the instructions
for authors of the Journal:
“In the text, reference numbers should be placed in square brackets [ ], and placed
before the punctuation; for example [1], [1–3] or [1,3]. For embedded citations in the
text with pagination, use both parentheses and brackets to indicate the reference number
and page numbers; for example [5] (p. 10). or [6] (pp. 101–105).
The reference list should include the full title, as recommended by the ACS style guide.
Style files for Endnote and Zotero are available.”

AR: Done

5) L42: Whenever a study is cited along the text, some background on the study
should be provided. It would be interesting if the authors could further elaborate
on which studies are these, which is their design, where are they from, and add
references accordingly.

AR: Done

6) L46-47: Which studies are these? Please provide further information.

AR: Done

7) L48: The authors could also mention the implications to herd immunity.

AR: Done

8) L49-50: Please provide appropriate references.

AR: Done

9) L56-58: Please provide references and further characterize numbers and
percentages. What is the frequency?

AR: Done

10) L63: The methods section should be improved. The authors should try to answer
these questions:
a) Why was this study protocol chosen? Was it based on previous literature?

Why this timeframe of 1 month? Please provide appropriate reference.

AR: The study protocol was based on previous literature (Berkovic et al., 2006). The timeframe of 1 month (exactly 4 weeks) comes from the Regulation of the Polish Minister of Health, reference added.

  1. b) Which study design type was this? Retrospective? Please specify “several projects”.

Was this a cohort study? If so, provide citations to the other studies involved.

AR: It was a descriptive case series study. The design of the study was ambidirectional, both retrospective (the analysis of existed medical documentation) and prospective (performing genetic tests and their analysis). The patients were collected and investigated during the course of Institute of Mother and Child (IMC) statutory projects or as a part of routine genetic diagnostics carried out in IMC.

Added to the Methods.

  1. c) During which years did the study take place?

AR: The study took place in 2019-2022.

Added to the Methods.

  1. d) Which was the setting of the study? Private practice? Academic hospital?

AR: The study was performed at the Institute of Mother and Child, a hospital which is a research and development unit of the Polish Ministry of Health.

Added to the Methods

  1. e) L67-68: How were these conditions diagnosed? Which were diagnostic criteria used?

Were appropriate specialists involved in their diagnoses? Were
EEG, video EEG, and neuroimaging performed in all of the patients with
seizures?

AR: Added to Methods

  1. f) A table with baseline characteristics of the participants should be included.

AR: Baseline chcaracteristics, such as age, sex, and ethnicity have been added to Supplementary Material 1.

  1. g) Were there following-up assessments of the patients? For how long?

AR: All patients included in the study remain under the care of multi-specialty reference centers. In the study group, 4 patients, due to significant disorders associated with developmental epileptic encephalopathy, require pediatric, neurological, rehabilitation, neuropsychological and speech therapy care. The families of the patients with molecular diagnosis were taken care of by the genetic clinic.

  1. h) How was the sample size calculated? Which was the power of the study?
    Which was the distribution? Which statistical analyzes were performed? It
    would be interesting if authors could provide multivariate analyzes of their
    data to improve the quality of the study. Were genetic factors related to
    clinical characteristics and vaccine outcomes? P-values and confidence
    intervals should be provided, as well as the parameters analyzed.

AR:

The case series study, which we present is not analytical, but descriptive one. Thus, it can not be used for hypothesis verification, but for hypothesis generation. Sample size is too small to perform statistical analyses, but the study, as we think, is good contribution to a study with a greater strength of evidence, i.e. analytical cohort study.

  1. i) Which criteria were used to classify drug-resistant epilepsy? Which AED
    were they on?

AR: The adopted criteria for drug-resistant epilepsy were in line with the current recommendations of the International League Against Epilepsy (ILAE) - revision 2020. Among the analyzed group, antiepileptic treatment included most drugs in accordance with the ILAE guidelines and the summary of product characteristic adjusted to the child's age, types of seizures or epilepsy syndrome.

11) Why was this genetic panel chosen? Was this decision based on previous
studies? Please provide references

AR: Panel was designed at the IMC based on previous studies. References added.

12) L175: Which dosage of ganaloxone was used? Why was this drug used? Which
drug presentation was chosen and why? For how long? How was drug response
objectively evaluated?

AR: In Patient 9, Ganaxolone was used at a dose of 50mg/kg/day for 12 weeks, improving the reduction of epileptic seizures. The drug was used according to clinical trials outcomes for patients with PCDH19 variants.

13) Further characterization of the epilepsy syndromes and seizures should be
provided, as well as the developmental delay features. Better characterization of
the neurological examination of the patients should also be provided, as well as
information on which was their frequency of following-up and which
professionals were following up them

AR: Added to Methods.

14) The authors should provide further clinical and epidemiological information
about the patients.

AR: Added do Methods.

15) Which type of patient population is this? Which is the age of epilepsy diagnosis?
Which drugs are they on? Were other AED tried? What is the frequency of the
seizures per day? Do they have more than one type of seizure? How many drugs
were tried? How many of them had status epilepticus? How many years of
following-up do these patients have? What was the number of seizures before
diagnosis? What countries are the patients from?

AR: Added to Methods.

16) Since when the following up took place? What was the ethnicity of the patients?
Were there neuroimaging abnormalities? What were the EEG and video-EEG
abnormalities? Did they have febrile seizures, or neonatal seizures, what was
their gestational age? Did they have a low weight at birth? The developmental
delay should be further characterized as well as neurological examination. What
was their seizure onset type? Do they have drug-resistant epilepsy? All of these
characteristics are relevant and should not be ignored.

AR: Added to Methods.

17) Was this study protocol based on previous studies? Why were these tests
performed? What were the researchers hoping to achieve with these tests? What
have they done differently from other studies? How does their research finding
differ from previously published literature? What does this research add of new?
How does this research contribute to clinical practice, and what are the practical
implications and the clinical relevance of this study?

AR: The answers to these questions were included in the text. The protocol was based on previous studies (Berkovic et al. 2006). These tests were performed in order to find a relation between AEFI and gene variants. We extended methods to perform whole exome sequencing, not only single gene analysis. To our knowledge, high-throughput genetic analyses performed on AEFI patients have never been published. We found new genes that may be related to AEFI. We showed that most of the genes have never been correlated with AEFI in scientific literature. Our findings let to adjust therapeutic management of AEFI patients according to molecular diagnosis we made (Ganaxolone and MAD). Our results show that more attention should be paid to the role of genetic factors in the pathogenesis of neurological AEFI and that all patients with such reactions should be referred for genetic examination, because the outcome may be of importance in therapeutic management.

18) Tables should be significantly improved and only include relevant information,
especially table 2.

AR: Done

19) The discussion should include limitations of the study such as the small sample
size and biases related to the study design type.

AR: Done

20) The authors should be careful with their conclusions because it doesn’t mean
that the correlations they found have a causal relationship. To even begin to
think of such a causal relationship as the authors imply in their results and
discussion, at minimum the authors should have performed multivariate
analyzes. The discussion seems to overreach. The authors should take care to not
be categorical in their affirmations and try only to “suggest” and add statements
such as” it is possible that, it is probable that, our study suggests” since they
have a significantly small sample-sized study.

AR: Comments were taken into account in the Discussion

21) The authors should review reporting guidelines for their study type and adequate
their introduction, abstract, methods, results, and discussion sections.

AR: Done

22) In general, authors should review the text and add all the relevant citations
through the text as many seem to be missing. Also, authors should work with
English editing services to improve the readability of the article.

AR: Missing 26 references have been added. The manuscript has been checked by English language specialist.

Round 2

Reviewer 2 Report

The authors greatly improved the article and there are only a few corrections left to be made.

Author Response

Reviewer 2_Review Report (Round 1)

Firstly, I would like to congratulate the authors on their interesting and relevant work.
However, there are some points that need to be addressed:

1) The title should contain the study design type.

Author’s response (AR): The title has been changed to:

Genetic risk factors for neurological disorders in children with adverse events following immunization. A descriptive study of Polish case series.

The authors should include the symbol “:” instead of “.” To separate the sentences in the title as the title cannot be fragmented.

AR: Done

2) The specific study design type should also be mentioned in the abstract.

AR: Done

3) I advise the authors to include standardized key-words according to MeSH terms
as this facilitates the search of the article in the future. Please see this website for
more information: https://www.ncbi.nlm.nih.gov/mesh/

AR: Done

4) L39: The authors should adequate the references according to the instructions
for authors of the Journal:
“In the text, reference numbers should be placed in square brackets [ ], and placed
before the punctuation; for example [1], [1–3] or [1,3]. For embedded citations in the
text with pagination, use both parentheses and brackets to indicate the reference number
and page numbers; for example [5] (p. 10). or [6] (pp. 101–105).
The reference list should include the full title, as recommended by the ACS style guide.
Style files for Endnote and Zotero are available.”

AR: Done

5) L42: Whenever a study is cited along the text, some background on the study
should be provided. It would be interesting if the authors could further elaborate
on which studies are these, which is their design, where are they from, and add
references accordingly.

AR: Done

On L52 the authors mention “systematic review studies” but provide only one citation.

AR: Corrected

6) L46-47: Which studies are these? Please provide further information.

AR: Done

7) L48: The authors could also mention the implications to herd immunity.

AR: Done

8) L49-50: Please provide appropriate references.

AR: Done

9) L56-58: Please provide references and further characterize numbers and
percentages. What is the frequency?

AR: Done

10) L63: The methods section should be improved. The authors should try to answer
these questions:
a) Why was this study protocol chosen? Was it based on previous literature?

Why this timeframe of 1 month? Please provide appropriate reference.

AR: The study protocol was based on previous literature (Berkovic et al., 2006). The timeframe of 1 month (exactly 4 weeks) comes from the Regulation of the Polish Minister of Health, reference added.

  1. b) Which study design type was this? Retrospective? Please specify “several projects”.

Was this a cohort study? If so, provide citations to the other studies involved.

AR: It was a descriptive case series study. The design of the study was ambidirectional, both retrospective (the analysis of existed medical documentation) and prospective (performing genetic tests and their analysis). The patients were collected and investigated during the course of Institute of Mother and Child (IMC) statutory projects or as a part of routine genetic diagnostics carried out in IMC.

Added to the Methods.

Please provide the Institutional Review Board Number of the study.

AR: Done

  1. c) During which years did the study take place?

AR: The study took place in 2019-2022.

Added to the Methods.

  1. d) Which was the setting of the study? Private practice? Academic hospital?

AR: The study was performed at the Institute of Mother and Child, a hospital which is a research and development unit of the Polish Ministry of Health.

Added to the Methods

  1. e) L67-68: How were these conditions diagnosed? Which were diagnostic criteria used?

Were appropriate specialists involved in their diagnoses? Were
EEG, video EEG, and neuroimaging performed in all of the patients with
seizures?

AR: Added to Methods

L122: Were at least 2 neurologists involved in the diagnosis of the patients?

AR: Yes, there were several independent doctors (pediatricians and child neurologists) involved in the diagnosis of the patients.

  1. f) A table with baseline characteristics of the participants should be included.

AR: Baseline chcaracteristics, such as age, sex, and ethnicity have been added to Supplementary Material 1.

  1. g) Were there following-up assessments of the patients? For how long?

AR: All patients included in the study remain under the care of multi-specialty reference centers. In the study group, 4 patients, due to significant disorders associated with developmental epileptic encephalopathy, require pediatric, neurological, rehabilitation, neuropsychological and speech therapy care. The families of the patients with molecular diagnosis were taken care of by the genetic clinic.

  1. h) How was the sample size calculated? Which was the power of the study?
    Which was the distribution? Which statistical analyzes were performed? It
    would be interesting if authors could provide multivariate analyzes of their
    data to improve the quality of the study. Were genetic factors related to
    clinical characteristics and vaccine outcomes? P-values and confidence
    intervals should be provided, as well as the parameters analyzed.

AR:

The case series study, which we present is not analytical, but descriptive one. Thus, it can not be used for hypothesis verification, but for hypothesis generation. Sample size is too small to perform statistical analyses, but the study, as we think, is good contribution to a study with a greater strength of evidence, i.e. analytical cohort study.

  1. i) Which criteria were used to classify drug-resistant epilepsy? Which AED
    were they on?

AR: The adopted criteria for drug-resistant epilepsy were in line with the current recommendations of the International League Against Epilepsy (ILAE) - revision 2020. Among the analyzed group, antiepileptic treatment included most drugs in accordance with the ILAE guidelines and the summary of product characteristic adjusted to the child's age, types of seizures or epilepsy syndrome.

11) Why was this genetic panel chosen? Was this decision based on previous
studies? Please provide references

AR: Panel was designed at the IMC based on previous studies. References added.

12) L175: Which dosage of ganaloxone was used? Why was this drug used? Which
drug presentation was chosen and why? For how long? How was drug response
objectively evaluated?

AR: In Patient 9, Ganaxolone was used at a dose of 50mg/kg/day for 12 weeks, improving the reduction of epileptic seizures. The drug was used according to clinical trials outcomes for patients with PCDH19 variants.

13) Further characterization of the epilepsy syndromes and seizures should be
provided, as well as the developmental delay features. Better characterization of
the neurological examination of the patients should also be provided, as well as
information on which was their frequency of following-up and which
professionals were following up them

AR: Added to Methods.

14) The authors should provide further clinical and epidemiological information
about the patients.

AR: Added do Methods.

15) Which type of patient population is this? Which is the age of epilepsy diagnosis?
Which drugs are they on? Were other AED tried? What is the frequency of the
seizures per day? Do they have more than one type of seizure? How many drugs
were tried? How many of them had status epilepticus? How many years of
following-up do these patients have? What was the number of seizures before
diagnosis? What countries are the patients from?

AR: Added to Methods.

16) Since when the following up took place? What was the ethnicity of the patients?
Were there neuroimaging abnormalities? What were the EEG and video-EEG
abnormalities? Did they have febrile seizures, or neonatal seizures, what was
their gestational age? Did they have a low weight at birth? The developmental
delay should be further characterized as well as neurological examination. What
was their seizure onset type? Do they have drug-resistant epilepsy? All of these
characteristics are relevant and should not be ignored.

AR: Added to Methods.

17) Was this study protocol based on previous studies? Why were these tests
performed? What were the researchers hoping to achieve with these tests? What
have they done differently from other studies? How does their research finding
differ from previously published literature? What does this research add of new?
How does this research contribute to clinical practice, and what are the practical
implications and the clinical relevance of this study?

AR: The answers to these questions were included in the text. The protocol was based on previous studies (Berkovic et al. 2006). These tests were performed in order to find a relation between AEFI and gene variants. We extended methods to perform whole exome sequencing, not only single gene analysis. To our knowledge, high-throughput genetic analyses performed on AEFI patients have never been published. We found new genes that may be related to AEFI. We showed that most of the genes have never been correlated with AEFI in scientific literature. Our findings let to adjust therapeutic management of AEFI patients according to molecular diagnosis we made (Ganaxolone and MAD). Our results show that more attention should be paid to the role of genetic factors in the pathogenesis of neurological AEFI and that all patients with such reactions should be referred for genetic examination, because the outcome may be of importance in therapeutic management.

18) Tables should be significantly improved and only include relevant information,
especially table 2.

AR: Done

19) The discussion should include limitations of the study such as the small sample
size and biases related to the study design type.

AR: Done

20) The authors should be careful with their conclusions because it doesn’t mean
that the correlations they found have a causal relationship. To even begin to
think of such a causal relationship as the authors imply in their results and
discussion, at minimum the authors should have performed multivariate
analyzes. The discussion seems to overreach. The authors should take care to not
be categorical in their affirmations and try only to “suggest” and add statements
such as” it is possible that, it is probable that, our study suggests” since they
have a significantly small sample-sized study.

AR: Comments were taken into account in the Discussion

21) The authors should review reporting guidelines for their study type and adequate
their introduction, abstract, methods, results, and discussion sections.

AR: Done

22) In general, authors should review the text and add all the relevant citations
through the text as many seem to be missing. Also, authors should work with
English editing services to improve the readability of the article.

AR: Missing 26 references have been added. The manuscript has been checked by English language specialist.